

# Benthic grazing in a eutrophic river: cascading effects of zoobenthivorous fish mask direct effects of herbivorous fish

Madlen Gerke[1], Daniel Cob Chaves[1], Marc Richter[1], Daniela Mewes[1], Jörg Schneider[2], Dirk Hübner[3] and Carola Winkelmann[1]

[1] Institute for Integrated Natural Sciences, University of Koblenz-Landau, Koblenz, Germany
[2] Bürogemeinschaft für fisch- und gewässerökologische Studien, Frankfurt, Germany
[3] Bürogemeinschaft für fisch- und gewässerökologische Studien, Marburg, Germany

## ABSTRACT

Benthic grazing strongly controls periphyton biomass. The question therefore arises whether benthic grazing could be used as a tool to reduce excessive growth of periphyton in nutrient-enriched rivers. Although benthic invertebrate grazers reduce the growth of periphyton, this is highly context dependent. Here we assessed whether the only obligate herbivorous fish in European rivers, the common nase (*Chondrostoma nasus* L.), is able to reduce periphyton biomass in a eutrophic river. We conducted three consecutive *in situ* experiments at low, intermediate and high densities of nase in the river using standard tiles on the river bottom naturally covered with periphyton that were accessible to fish and tiles that excluded fish foraging with electric exclosures. The biomass of benthic invertebrate grazers was very low relative to nase. We hypothesised that nase would reduce periphyton biomass on accessible tiles and therefore expected higher periphyton biomass on the exclosure tiles, at least at intermediate and high densities of nase in the river. Contrary to our expectation, the impact of fish grazing was low even at high fish density, as judged by the significantly lower chlorophyll *a* concentration on exclosure tiles even though the ash-free dry mass on accessible and exclosure tiles did not differ. The lower chlorophyll *a* concentrations on exclosure tiles might be explained by a higher biomass of invertebrate grazers on the exclosure tiles, which would indicate that the effect of invertebrate grazers was stronger than that of herbivorous fish grazers. The high biomass of invertebrate grazers on exclosure tiles likely arose from the exclusion of zoobenthivorous fish, which occur in the river at high densities. The results of our small-scale experiments suggested that cascading top-down effects of zoobenthivorous fish have a higher impact on periphyton biomass than direct effects of herbivorous nase.

# INTRODUCTION

Benthic grazing in running waters strongly affects periphyton biomass (*Feminella & Hawkins, 1995*; *Hillebrand, 2009*; *Holomuzki, Feminella & Power, 2010*). Invertebrates across different taxonomic groups are able to reduce standing crops of stream periphyton

Corresponding author
Madlen Gerke,
mgerke@uni-koblenz.de

(e.g., Gastropoda: *Rosemond, Mulholland & Elwood, 1993*; *Rosemond, Mulholland & Brawley, 2000*, Ephemeroptera: *Hill & Knight, 1987*; *Moulton et al., 2004*; Trichoptera: *Lamberti & Resh, 1983*; *Katano et al., 2007*), and benthic grazing limits periphyton biomass accrual in nutrient-enriched rivers (*Peterson et al., 1993*; *Sturt, Jansen & Harrison, 2011*). Herbivorous fish might also affect periphyton accumulation in streams and shallow rivers. Strong top-down effects of herbivorous fish have been found in subtropical (*Schneck, Schwarzbold & Melo, 2013*) and tropical streams (*Power, Dudley & Cooper, 1989*; *Wootton & Oemke, 1992*; *Flecker et al., 2002*). However, it has been suggested that fish grazing is more important in tropical streams than in temperate streams due to the higher density and species richness of herbivorous fish in the tropics (*Wootton & Oemke, 1992*). Nevertheless, field experiments have shown that the highly abundant small herbivorous cyprinid *Campostoma anomalum* reduces periphyton biomass in North American streams (*Power, Matthews & Stewart, 1985*; *Stewart, 1987*; *Gelwick & Matthews, 1992*). In mesocosms simulating small headwater prairie streams, the presence of one or two herbivorous fish species (*Chrosomus erythrogaster, C. anomalum*) results in a reduction of algal filament lengths and periphyton biomass (*Martin et al., 2016*).

In the light of the potentially high top-down impact of grazers in streams and rivers, the question arises whether enhancement of benthic grazing could be used as a tool for the mitigation of eutrophication effects in stream conservation approaches, similar to biomanipulation in lakes (*Shapiro & Wright, 1984*; *Hansson et al., 1998*). In streams and shallow rivers, nutrient enrichment promotes excessive growth of periphyton, which in turn can cause high diurnal fluctuations of oxygen and pH (*Dodds & Welch, 2000*; *Hilton et al., 2006*) and biological clogging of the hyporheic zone (*Ibisch, Seydell & Borchardt, 2009*). This reduces the habitat quality for fish and invertebrates (*Welch, Quinn & Hickey, 1992*; *Hübner, Borchardt & Fischer, 2009*).

However, although the effects of invertebrate grazers on stream periphyton are generally strong (*Hillebrand, 2009*), they do not seem to be sufficient to prevent algal blooms owing to a temporal mismatch of algal and invertebrate generation times (*Rosemond, Mulholland & Brawley, 2000*; *Winkelmann et al., 2014*). This mismatch has the highest effect in small streams in forested catchments, when canopy cover strongly limits periphyton growth during the vegetation season and therefore algal blooms occur usually only in spring prior to tree foliation (*Rosemond, Mulholland & Brawley, 2000*; *Winkelmann et al., 2014*). In rivers, however, a full canopy cover is rarely reached; therefore, periphyton are not greatly light limited during the vegetation season. Hence, the seasonal offset of algal production and grazing pressure observed in narrow streams might be not as strong in wider rivers. Furthermore, rivers also accommodate larger and often more diverse fish communities, including herbivorous or facultative herbivorous fish (*Oberdorff, Guilbert & Lucchetta, 1993*). Consequently, in shallow rivers, a top-down control of periphyton might be facilitated by the promotion of fish grazing, possibly in combination with the enhancement of invertebrate grazing.

In European rivers, the large cyprinid common nase (*Chondrostoma nasus* L.) is the only obligate herbivorous fish species (*Vater, 1997*). It feeds exclusively on periphyton, and preferentially on benthic diatoms (*Freyhof, 1995*; *Corse et al., 2010*). Despite large-scale

population declines, nase is still abundant in many European rivers (*Reckendorfer et al., 2001*; *Melcher, Lautsch & Schmutz, 2012*). The home ranges of nase are well defined, with an average daily activity range of 120 m (*Huber & Kirchhofer, 1998*); therefore, a continuous impact of nase on periphyton in these home ranges might be expected. However, to our knowledge, the quantitative impact of herbivorous nase on periphyton biomass has not yet been investigated. Therefore, our study aimed at assessing whether nase is able to reduce periphyton biomass in a eutrophic river. In three *in situ* experiments using standardized concrete tiles covered with periphyton and electrical exclusion, we quantified the effects of fish exclusion on periphyton biomass at different densities of nase. We expected that nase, as the only herbivorous fish in the river, would control periphyton biomass, i.e., that periphyton biomass would be lower on tiles accessible to fish than on exclosure tiles, at least at intermediate and high densities of nase.

## METHODS

### Experimental site

Experiments were conducted in the hyporhithral zone of the river Nister (Rhineland-Palatinate, Germany, 50°43′N, 7°44′E), a small gravel-bed river with a drainage area of 246 km$^2$. The average mean discharge is 6.4 m$^3$ s$^{-1}$ in winter and 2.4 m$^3$ s$^{-1}$ in summer (measured at Heimborn, ID 2724030100; data supplied by State Office for Environment of Rhineland-Palatinate). At the experimental sites, the river is about 10 m wide and never completely shaded during the vegetation season. The river bed mainly consists of cobbles (6.3–20 cm) and boulders (20–63 cm). Land use in the catchment is dominated by forestry, pasture and agriculture. Due to phosphate emissions from several minor municipal wastewater treatment plants and diffuse emissions from agriculture, nutrient levels in the river are high (mean ± SD: 106 ± 62 μg PO$_4$-P L$^{-1}$, 5.3 ± 1.2 mg NO$_3$-N L$^{-1}$; $n = 18$; monthly measurements between June 2015 and July 2017, except during winter flood). Eutrophication effects, such as oxygen oversaturation and extreme pH, have been observed, especially during spring algal bloom (maximum in April 2016: 182.3% O$_2$, pH 10.2).

The benthic algal and cyanobacterial community in the river is largely composed of adnate and loosely attached diatoms. During summer, filamentous cyanobacteria or filamentous green algae, especially *Cladophora* spp., can become dominant. We conducted the experiments after the spring peak and breakdown of periphyton biomass in early summer.

Common nase is the only herbivorous fish in the river. The fish scrape periphyton from coarse substrate, typically swim in shoals and have defined home ranges. Fish grazing pressure in the river can be expected to be highest in run segments with coarse substrate, which are the preferred feeding habitats of nase (*Huber & Kirchhofer, 1998*). The invertebrate community is dominated by scraping grazers, especially mayfly larvae (*Baetis* spp., *Ephemerella ignita*), chironomid larvae and the snail *Ancylus fluviatilis*.

To quantify effects of fish exclusion, we conducted three consecutive experiments at three different densities of nase (Table 1). The first two experiments were performed in summer 2013 at two different sites (sites A and B) representing typical nase feeding habitats (20 m in

**Table 1  Fish densities in the river in the three experiments.** Experiment I = low nase density; experiment II = intermediate nase density, and experiment III = high nase density. Values are the total number of individuals caught per m$^2$ and calculated stock per m$^2$ (large fish: *Ricker, 1975*; small fish: *De Lury, 1951*, given only in case of significant regression coefficients).

| | Experiment I | | Experiment II | | Experiment III | |
| | Site A | | Site B | | Site B | |
| | July 2013 | | June 2013 | | July 2016 | |
| Fish | Catch (ind m$^{-2}$) | Stock (ind m$^{-2}$) | Catch (ind m$^{-2}$) | Stock (ind m$^{-2}$) | Catch (ind m$^{-2}$) | Stock (ind m$^{-2}$) |
|---|---|---|---|---|---|---|
| >15 cm | | | | | | |
| Nase | 0.005 | 0.004 | 0.019 | 0.033 | 0.198 | 0.276 |
| Chub | 0.003 | 0.005 | 0.003 | 0.005 | 0.032 | 0.062 |
| Dace | 0.004 | 0.005 | 0.003 | 0.010 | 0.016 | 0.082 |
| Other | 0.005 | 0.012 | 0.003 | 0.007 | 0.036 | 0.076 |
| Total | 0.016 | 0.026 | 0.027 | 0.055 | 0.281 | 0.495 |
| <15 cm | | | | | | |
| Bullhead | 0.51 | | 0.31 | 0.84 | 0.27 | 0.4 |
| Minnow | 0.48 | | 0.68 | | 0.25 | 0.4 |
| Stone loach | 0.85 | 1.23 | 0.52 | 1.49 | 0.91 | 2.0 |
| Total | 1.83 | | 1.51 | | 1.43 | 2.8 |

length) at low (experiment I) and intermediate (experiment II) densities. Site A (low nase density) is located 2.5 km upstream from site B (intermediate nase density), and nase density was eight times higher at site B than at site A (Table 1). In 2015, stocks of herbivorous nase and the omnivorous European chub (*Squalius cephalus*) were experimentally increased in a 500 m reach including site B for a long-term food web manipulation experiment; the experimental reach was defined by fish barriers to avoid fish emigration. In July 2016, we performed a third experiment (experiment III) at site B with a nase density more than 8-fold higher than in experiment II and 70-fold higher than in experiment I (Table 1). At the time of experiment III, nase biomass per area at site B was approximately 100-fold higher than the total benthic invertebrate biomass (nase: 111.5 g m$^{-2}$, total invertebrates: 1.2 g m$^{-2}$). Aside from nase, the omnivorous European chub and common dace (*Leuciscus leuciscus*) were the most abundant large fish at the experimental sites (Table 1). The small zoobenthivorous fish species bullhead (*Cottus gobio*), common minnow (*Phoxinus phoxinus*) and stone loach (*Barbatula barbatula*) generally occurred at high abundances at both experimental sites and during all experiments (Table 1). Fish stocks were assessed by electrofishing campaigns (EFGI 650, Bretschneider Spezialelektronik, Chemnitz, Germany) in June 2013 (site A and B) and July 2016 (site B). Electrofishing was approved by the fisheries department of the local environmental agency SGD Nord (Rhineland-Palatinate, Germany). For each experiment, stocks of large fish (>15 cm) were estimated in a 500-m reach including the experimental site using the mark-recapture method. Stocks of small fish (<15 cm) were estimated in 60-m (experiments I and II) and 40-m (experiment III) long sections close to the experimental site by a three-pass removal method.

## Electric exclosures

We used low-intensity electric pulses following the principle developed by *Pringle & Blake (1994)* to prevent fish but not benthic invertebrates from foraging on standardized concrete tiles (40 cm × 40 cm) exposed on the stream bottom. The strength of the electric field determines which organisms are affected by electrical exclusion because the sensitivity to electric fields increases with body size (*Moulton et al., 2004*). The electric field strength that we used in our experiments was comparable to that used in other field studies in which macroconsumers (approximately ≥1 cm, in this case fish and shrimps) were selectively excluded, while smaller invertebrates were not affected by the electric field (*Pringle & Blake, 1994*; *Pringle & Hamazaki, 1997*; *Rosemond, Pringle & Ramirez, 1998*). Electrical exclusion has the advantage that it avoids experimental artefacts associated with traditional enclosures or exclosures, such as reduced current velocity, increased sedimentation and shading effects.

Two protruding aluminium conductors were attached at opposite sides of each tile and were insulated at the bottom of the tile. Fish exclosure tiles were connected to commercially available electrical fence chargers (experiments I and II: compact B400, Electra Landtechnik GmbH, Vöhl, Germany; experiment III: Voss.farming Aures 3, Elefant-Weidezaungeräte e.K., Ohrstedt, Germany; both approximately 0.3 J output energy) that emitted approximately 50 electrical pulses per minute and were powered by a 12-V battery. Sets of three exclosures were connected in parallel to a fence charger. Control tiles were constructed in the same manner as exclosures but were not connected to a fence charger. The effectiveness of exclusion was tested in two preliminary experiments with five individuals of nase in artificial indoor-stream channels (2.6 m × 0.9 m × 0.5 m) at the University of Koblenz–Landau, Koblenz. In both 24-h experiments, electric pulses effectively prevented fish from foraging on electrified tiles but did not significantly unsettle the animals (see Article S1). This is consistent with observations that we made during the field experiments.

## Experimental setup and sampling

Experiments I (June 2013) and II (July 2013) ran 18 days, and experiment III (July 2016) ran 19 days. Each separate experiment had a total of 18 tiles; half the tiles were electrified to exclude fish, and the other half were non-electrified to allow fish access. Electrified and non-electrified tiles were placed at least 2 m apart from each other to avoid an electric field between exclosures and controls. To allow initial periphyton growth, tiles were exposed on the river bottom two weeks prior to the start of experiments. On at least three occasions during the experiments, we measured water depth (only in experiments I and II), photosynthetically active radiation (PAR) using a LI-250 light meter (LI-COR, Lincoln, NE, USA) with a spherical micro-quantum sensor (US-SWS/L; Heinz Walz GmbH, Effeltrich, Germany) and current velocity using a flow meter with a vane wheel flow sensor (HFA hand-held unit with FA sensor; Hoentzsch GmbH, Waiblingen, Germany) above each tile.

At the end of each experiment, we sampled periphyton and benthic invertebrates to control for possible effects of invertebrate grazing. Fence chargers were not turned off until immediately before sampling to ensure continuous fish exclusion. Periphyton and benthic

invertebrate samples were each taken from half of the area of each tile. First, half of the tile was covered with a metal frame (20 cm × 40 cm) to protect the area for periphyton sampling. Invertebrates were sampled from the uncovered area by scraping with a coarse brush. Animals and organic material were washed into a net (500-$\mu$m mesh), which was positioned at the downstream edge of the tile. Subsequently, the invertebrate samples were rinsed over a 500-$\mu$m sieve and stored in 70% ethanol. The tile was then carefully removed from the river bottom, and periphyton was removed from the other half of the tile by brushing the area carefully with a coarse brush and up to 500 mL river water. The resulting periphyton suspensions were transported in the dark to the laboratory.

In experiment III, additional periphyton and benthic invertebrate samples were taken approximately 10 m downstream of the experimental site to compare the colonization of tiles and natural substrates in the river. These samples were actually collected for another field study, and periphyton were sampled five days before the end of experiment III and invertebrates were sampled one day after the end of experiment III. Ten stones were randomly chosen over the entire width of the river in order to obtain one mixed sample; periphyton was removed by carefully brushing the stone surface with a coarse brush and river water. Benthic invertebrates were sampled with three Surber samplers (total area 0.24 m$^2$, 500-$\mu$m mesh).

## Laboratory analyses

Nutrient concentrations were measured photometrically (nitrate: *DIN EN ISOIS 13395, 1996*; phosphate: *DIN EN ISO 15681-2, 2005*) using a continuous flow analyser (CFA, AutoAnalyser 3; Seal Analytical GmbH, Norderstedt, Germany). Total periphyton biomass was estimated as ash-free dry mass (mg AFDM cm$^{-2}$), and autotrophic periphyton biomass was estimated as chlorophyll *a* concentration ($\mu$g Chl *a* cm$^{-2}$). Periphyton biomass was quantified considering the total volume of the obtained periphyton suspension and the sampled area of tiles (experiments I–III) and stones (additional samples in experiment III). The surface area of stones sampled close to the experimental site were determined by carefully wrapping the stone in aluminium foil; overlapping areas were cut off, and the foil was weighed. The total volume of each periphyton suspension was determined, and the suspension was then homogenized using a magnetic stirrer to ensure comparable aliquots. For quantification of AFDM, 10 mL aliquots were transferred to pre-weighed ceramic crucibles and dried at 60 °C for 24 h. Dried samples were weighed, ashed at 510 °C for 5 h in a muffle furnace and subsequently reweighed.

To determine Chl *a* concentrations, triplicate aliquots were centrifuged at 13,000 rpm for 3 min (16,060 ×*g*, Micro 200R; Hettich Zentrifugen, Tuttlingen, Germany). The aliquot volume was 2 mL in experiments I and II but 0.5 mL in experiment III because the periphyton suspensions were thicker. The supernatants were discarded, and pellets were stored at −80 °C . Chl *a* was extracted and spectrophotometrically analysed according to *Mewes, Spielvogel & Winkelmann (2017)*. In short, pellets were homogenized in 500 $\mu$L of 96% ethanol buffered with 1 g MgCO$_3$ L$^{-1}$ using a disperser (Ultra Turrax T8; IKA, Staufen, Germany), except for samples in experiment III, which were homogenized in a mixing mill (MM 400; Retsch Technology GmbH, Haan, Germany). Another 0.5, 1.0 or 1.5 mL

of buffered 96% ethanol was added, depending on the intensity of green colouration, to prevent incomplete extraction in high-quantity samples. Chl $a$ was extracted for at least 3 h at room temperature in the dark. Subsequently, the samples were centrifuged at 6,000 rpm (3,421 $\times g$, Micro 200R) for 3 min, and Chl $a$ in the supernatant was measured spectrophotometrically (Specord 205; Analytic Jena, Jena, Germany) at 665 nm and corrected for turbidity at 750 nm. If the sample absorbance exceeded 1, the sample was appropriately diluted with buffered ethanol to give a reading of less than 1.

In experiment III, additional 2 mL aliquots were taken from the homogenized periphyton suspensions and stored at $-80\,°C$ for later analysis of the benthic algal and cyanobacterial community composition. The mean percentage of each taxonomic group (diatoms, green algae and cyanobacteria) in the suspension was estimated microscopically (400 × magnification) relative to the total area covered by algae and cyanobacteria on the slide (which was set to 100% in each microscopic field of view). The mean percentage of each group was estimated from 50 fields of view per slide, and three slides were analysed per sample.

All individuals from each benthic macroinvertebrate sample were sorted under a dissecting microscope, identified to the lowest practicable taxonomic level and counted. For each taxon in each sample, at least 50 individuals were measured to the nearest 0.1 mm. If less than 50 individuals occurred for one taxon per sample, all individuals of the sample were measured. The individual body mass (dry mass) was calculated using length–weight regressions. For all taxa except Chironomidae, we used regression models from the literature (*Meyer, 1989*; *Benke et al., 1999*; *Baumgärtner & Rothhaupt, 2003*; *Edwards et al., 2009*). For Chironomidae, we used data from our own samples to obtain a power function relating body length (BL) and dry mass (DM): $DM = 0.0013*BL^{2.8024}$ ($r^2 = 0.96$, $n = 62$). We did this because Chironomidae were the dominant group in most of our samples, they consist of species different than those in samples reported in literature, and published regressions for Chironomidae are based on a relatively small sample size (e.g., 16 samples in *Meyer, 1989*). The samples used to determine the body length and dry mass of chironomid larvae were taken from the river Nister and had been stored in 70% ethanol for 6 months. The length of undamaged individuals was measured to the nearest 0.1 mm. Afterwards, each individual was transferred to a pre-weighed reaction tube and dried for 24 h at 60 °C. After cooling in a desiccator, dry mass was determined to the nearest 0.01 mg using a microbalance (XS205 Dual Range; Mettler-Toledo, Columbus, OH, USA). To reduce measurement error, the dry mass of smaller specimens (<5 mm) was determined by weighing 2–15 individuals of a similar length together and calculating a mean individual body mass. To correct the individual dry mass for mass loss owing to preservation, we used a conversion factor of 1.26 (*Mährlein et al., 2016*).

## Data analysis

Stocks of large fish (>15 cm) were calculated using Chapman's modified Petersen estimator (*Ricker, 1975*). Stocks of small fish (<15 cm) were calculated using the *De Lury (1951)* regression method. In the case of non-significant regression coefficients ($R^2 < 0.88$), only the total number of caught individuals per m$^2$ is given because it represents a minimum

estimation for the population density of small fish. Chl $a$ concentration was calculated per area of the tile surface (as in *Mewes, Spielvogel & Winkelmann, 2017*). Means of periphyton biomass (Chl $a$ and AFDM) on exclosure and accessible tiles were compared using independent $t$-tests.

We assessed differences in the invertebrate community composition between exclosure and accessible tiles using analysis of similarities (ANOSIM) based on benthic invertebrate biomass. If ANOSIM results were significant, a similarity percentage (SIMPER) analysis was used to identify the taxa that were mainly responsible for the differences between exclosure and accessible tiles. In experiments I and III, nine exclosure and nine accessible tiles were analysed; in experiment II, only eight tiles of each type were analysed due to desiccation of one sample. To calculate the total biomass of invertebrate grazers in each sample, we weighted the biomass of each herbivorous and omnivorous species according to the average proportion of plant food in their diet (*Schmedtje & Colling, 1996*). The mean total grazer biomass of benthic invertebrates and the mean biomass of several dominant grazer taxa on exclosure and accessible tiles were compared using $t$-tests and adjusted for multiple comparisons using the Bonferroni-Holm correction.

To check whether environmental factors distorted the effects of fish exclusion on periphyton biomass, we used $t$-tests of differences in mean water depth, PAR and current velocity (averaged over the experimental period for each tile) between exclosure and accessible tiles. In addition, Pearson correlations were calculated to assess any influence of environmental factors on periphyton and grazer biomass. Non-normally distributed data were log-transformed for Pearson's correlation analysis. For all comparisons of means, Welch's test was used instead of the $t$-test when the assumption of homogeneity of variance was not met. Statistical analyses were performed and graphs were plotted using R version 3.3.3 (*R Development Core Team, 2016*).

## RESULTS

When fish were excluded from the tiles, autotrophic periphyton biomass decreased, as shown by the significantly lower Chl $a$ levels on tiles inaccessible to fish (exclosure) than on the tiles to which fish had access (control) at intermediate (experiment II) and high (experiment III) densities of nase (II: $p < 0.01$; III: $p = 0.02$; $n = 9$; $t$-test; Figs. 1B and 1C). At a low nase density (experiment I), Chl $a$ concentrations on the fish exclosure tiles were not significantly lower (Chl $a$: $p = 0.06$, $n = 9$, $t$-test, Fig. 1A). By contrast, total periphyton biomass measured as AFDM did not differ between accessible and exclosure tiles (I: $p = 0.33$, Welch test; II: $p = 0.19$, $t$-test; III: $p = 0.93$, Welch test; $n = 9$, Figs. 2A–2C). At a high nase density (experiment III), the variance of AFDM was significantly higher on the exclosure tiles than on the accessible tiles ($p < 0.01$, $n = 9$, $F$-test, Fig. 2C). Overall, both the Chl $a$ concentration and AFDM were lowest on both sets of tiles at a low nase density and highest at a high nase density.

The algal and cyanobacterial communities on the tiles at a high nase density were dominated by diatoms (mean $\pm$ SD, $n = 9$: exclosures: 63% $\pm$ 9%, controls: 59% $\pm$ 6%), followed by green algae (exclosures: 33% $\pm$ 9%, controls: 27% $\pm$ 8%) and cyanobacteria
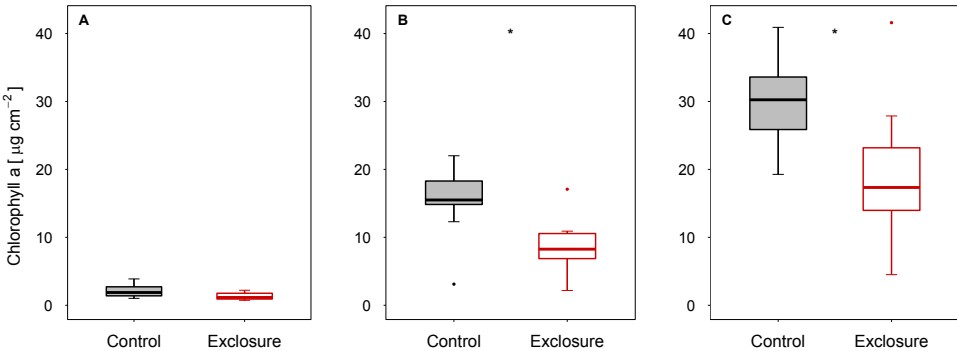

**Figure 1  Chlorophyll *a* concentration for controls and exclosures at the end of the experiments.** Chlorophyll *a* per area on accessible control tiles and fish exclosure tiles ($n = 9$) at the end of the three experiments performed at (A) low, (B) intermediate and (C) high densities of nase. Boxes: 75 and 25%, whiskers: 95 and 5%, dots: outliers. * Significant ($p < 0.05$).

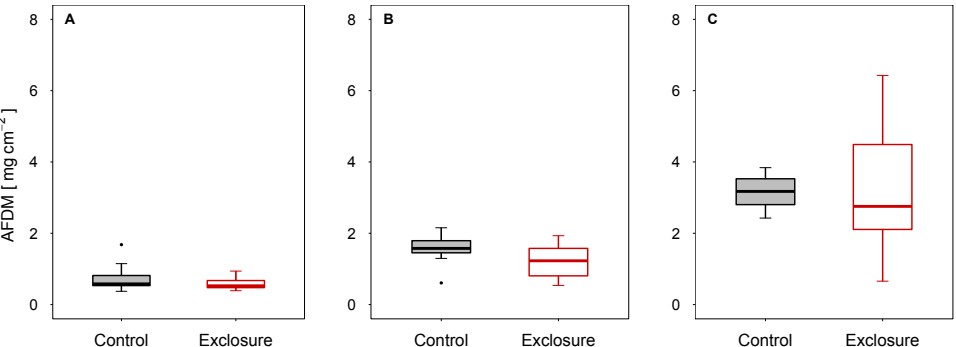

**Figure 2  Ash-free dry mass for controls and exclosures at the end of the experiments.** Ash-free dry mass per area on accessible control tiles and fish exclosure tiles ($n = 9$) at the end of the three experiments performed at (A) low, (B) intermediate and (C) high densities of nase. Boxes: 75 and 25%, whiskers: 95 and 5%, dots: outliers.

(exclosures: 3% ± 4%, controls: 14% ± 10%). Green algae were mostly filamentous, and included the taxa *Cladophora* spp. and *Microspora* spp. Benthic diatoms included loosely attached taxa such as *Navicula* spp., and stalked taxa such as *Gomphonema* spp., which were often found epiphytic on *Cladophora* spp. Cyanobacteria included mostly filamentous taxa, especially *Phormidium* spp. and bundles of *Homoeothrix* spp. The estimated proportion of cyanobacteria was higher on the accessible control tiles than on the exclosure tiles ($p < 0.01$, $n = 9$, $t$-test), which might point to effects of grazing on the periphyton community structure. On natural river stones, the algal and cyanobacterial communities differed from those on the tiles (mixed sample obtained from ten stones: 60% filamentous cyanobacteria, 38% diatoms, 2% filamentous green algae). The difference between tiles and natural substrates might have been even more intense owing to the time delay between samplings especially because there was a sunny period between the sampling dates. However, the communities on natural river stones and on the tiles generally consisted of similar taxa

(e.g., *Homoeothrix* spp., *Phormidium* spp., *Navicula* spp., *Gomphonema* spp., *Microspora* spp.). Periphyton biomass on the river stones was in the same order of magnitude as on the accessible tiles (mixed sample obtained from ten stones: 23.7 µg cm$^{-2}$ Chl *a*, 2.4 mg cm$^{-2}$ AFDM; accessible tiles of experiment III (mean $\pm$ SD, $n = 9$): $30.0 \pm 6.5$ µg cm$^{-2}$ Chl *a*, $3.1 \pm 0.5$ mg cm$^{-2}$ AFDM).

The composition of the benthic invertebrate community differed between exclosure and accessible tiles at high and intermediate nase densities, but not at a low nase density (I: $R = 0.08$, $p = 0.13$; II: $R = 0.18$, $p = 0.03$; III: $R = 0.64$, $p = 0.001$, ANOSIM). At a high nase density (experiment III), Chironomidae and *Baetis* spp. contributed most to the dissimilarity between exclosure and accessible tiles (34% and 28%, respectively; SIMPER). At an intermediate nase density (experiment II), Chironomidae and *Ephemerella ignita* contributed most to the dissimilarity between exclosure and accessible tiles (31% and 18%, respectively, SIMPER). The biomasses of the taxa responsible for the differences in SIMPER were consistently higher on the fish exclosure tiles. In all three experiments, Chironomidae and mayfly grazers (*Baetis* spp., *Ephemerella ignita* and occasionally *Ecdyonurus* spp.) contributed most to grazer biomass.

Grazer biomass was affected by the presence of fish. At a high nase density (experiment III), total grazer biomass and biomass of mayfly grazers were significantly higher on the fish exclosure tiles than on the accessible control tiles (total: $p < 0.001$, mayfly: $p = 0.001$; chironomid: $p = 0.16$, $n = 9$; Welch test; Fig. 3C). At an intermediate nase density, at least the Chironomidae biomass showed a tendency to increase (II: total: $p = 0.16$, mayfly: $p = 0.16$, chironomid: $p = 0.051$; $n = 8$; Welch test; Fig. 3B). At a low nase density, grazer biomass did not differ between exclosure and accessible tiles (I: total: $p = 0.30$, mayfly: $p = 0.24$, chironomid: $p = 0.36$; $n = 9$; *t*- test, Fig. 3A), and grazer biomass was overall lowest in this experiment.

Invertebrate colonization of the tiles was comparable but not identical to that of the natural substrates close to the experimental site in experiment III. Some taxa, such as *Ancylus fluviatilis* and *Elmis* sp., occurred more often on natural substrates. However, Chironomidae, *Baetis* spp. and *Ephemerella ignita* were among the most important invertebrate grazers (45% of grazer biomass) in the river, and had densities comparable to those on the tiles (see Table S2).

There was no evidence that the environmental factors water depth, light and current velocity affected the experimental results. The water depths at exclusion and control tile sites were similar (I: $p = 0.82$; II: $p = 0.75$; III not measured; *t*-test; $n = 9$; Table 2). The light supply was also similar at exclusion and control tile sites in all three experiments (I: $p = 0.72$, Welch test; II: $p = 0.64$, *t*-test; III: $p = 0.45$, Welch test; $n = 9$, Table 2). Current velocities were similar at exclusion and control tile sites in experiments I and II (I: $p = 0.88$, II: $p = 0.63$, $n = 9$, *t*-test) but was significantly higher around the fish exclosure tiles than around the control tiles in experiment III ($p = 0.03$, $n = 9$, *t*-test, Table 2). However, although some significant correlations were found between environmental factors and biotic response variables in experiments I and II (Table 3), there was no significant

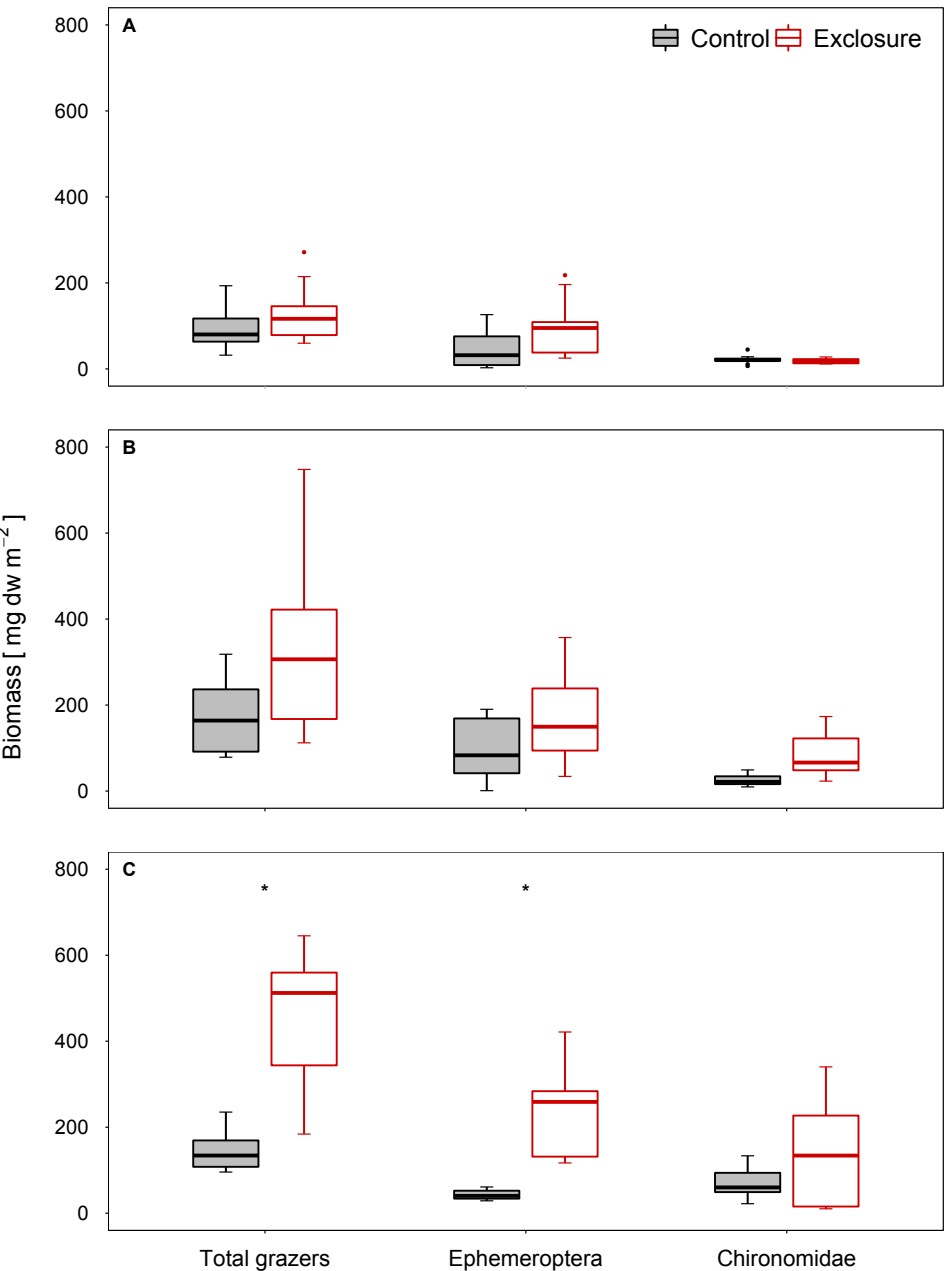

**Figure 3 Biomass of invertebrate grazers for controls and exclosures at the end of the experiments.**
Biomass of invertebrate grazers on accessible control tiles and fish exclosure tiles ($n \geq 8$) at the end of the three experiments performed at (A) low, (B) intermediate and (C) high densities of nase. Boxes: 75 and 25%, whiskers: 95 and 5%, dots: outliers. * Significant ($p < 0.05$).

**Table 2  Mean values of environmental parameters for controls and exclosures in the three experiments.** Mean values ($\pm$ SD) of water depth, light supply (PAR) and current velocity on accessible control tiles and fish exclosure tiles ($n = 9$) averaged over the experiment duration (number of measurements, $n \geq 3$).

| Nase density | Water depth (cm) | | PAR ($\mu$mol m$^{-2}$ s$^{-1}$) | | Current velocity (m s$^{-1}$) | |
|---|---|---|---|---|---|---|
| | Exclosure | Control | Exclosure | Control | Exclosure | Control |
| Low (I) | $21.9 \pm 10.3$ | $20.7 \pm 11.5$ | $846.1 \pm 696.2$ | $979.1 \pm 849.1$ | $0.19 \pm 0.04$ | $0.20 \pm 0.08$ |
| Intermediate (II) | $29.3 \pm 13.5$ | $27.5 \pm 9.8$ | $153.6 \pm 61.7$ | $169.4 \pm 78.0$ | $0.27 \pm 0.12$ | $0.29 \pm 0.08$ |
| High (III) | | | $127.4 \pm 45.3$ | $111.8 \pm 41.4$ | $0.33 \pm 0.06$ | $0.27 \pm 0.06$ |

**Table 3  Pearson correlation coefficients between biotic response variables and environmental factors (water depth, light supply PAR and current velocity).**

| | Low nase density (I) | | | Intermediate nase density (II) | | | High nase density (III) | |
|---|---|---|---|---|---|---|---|---|
| | Water depth | PAR | Current velocity | Water depth | PAR | Current velocity | PAR | Current velocity |
| **Exclosure** | | | | | | | | |
| Chl a | 0.89[*] | −0.31 | 0.58 | 0.02 | −0.36 | 0.75[*] | 0.60 | 0.24 |
| AFDM | 0.62 | 0.03 | 0.60 | −0.07 | −0.12 | 0.72[*] | 0.62 | 0.23 |
| Grazer biomass | −0.36 | 0.75[*] | 0.44 | −0.57 | 0.48 | 0.58 | 0.57 | 0.18 |
| **Control** | | | | | | | | |
| Chl a | 0.72[*] | −0.15 | 0.50 | −0.23 | 0.44 | −0.22 | 0.45 | 0.51 |
| AFDM | 0.64 | 0.03 | 0.51 | −0.03 | 0.21 | −0.36 | 0.44 | 0.44 |
| Grazer biomass | −0.83[*] | 0.68[*] | 0.73[*] | −0.32 | 0.41 | 0.40 | −0.02 | 0.48 |

Notes.
[*]Significant ($p < 0.05$).

relationship between current velocity and Chl a (exclosures: $p = 0.45$; controls: $p = 0.18$; $n = 9$), AFDM (exclosures: $p = 0.47$; controls: $p = 0.22$; $n = 9$) or between current velocity and grazer biomass (exclosures: $p = 0.65$; controls: $p = 0.15$; $n = 9$; Table 3) in experiment III, where current velocity differed between exclusion and control tile sites.

## DISCUSSION

As grazers are able to strongly control periphyton biomass (*Feminella & Hawkins, 1995*; *Hillebrand, 2009*) and eutrophication-driven algal blooms detrimentally affect the ecological quality of running waters (*Dodds & Welch, 2000*; *Biggs, 2000*; *Hilton et al., 2006*), the active promotion of benthic grazing might be a future tool for improving the quality of shallow rivers or unshaded streams. However, in contrast to the known top-down effects of benthic invertebrate grazers, there is a considerable gap of knowledge concerning the top-down effects of herbivorous fish. To assess whether top-down effects of herbivorous fish are able to control periphyton biomass, we conducted *in situ* exclosure experiments in which fish were not allowed to gain access to tiles colonized by periphyton and compared the results to those obtained with accessible control tiles. The unspecific exclusion of fish in our study was expected to be sufficient to reduce grazing intensity, because the common nase (*C. nasus*) is a specialized periphyton feeder (*Freyhof, 1995*; *Corse et al., 2010*) and the only herbivorous fish in the river. We expected strong top-down effects by nase on periphyton biomass, especially because we frequently observed visible traces of feeding

of nase in the river during summer. Therefore we hypothesized that nase can reduce periphyton biomass, which in our experiments would result in higher periphyton biomass on exclosure tiles compared to accessible tiles in a eutrophic river with intermediate and high densities of nase.

In contrast to our hypothesis, autotrophic periphyton biomass (measured as Chl *a*) was lower on the exclosure tiles at intermediate and high densities of nase. This possibly indicates a low impact of fish grazing at this experimental scale. This result was surprising because it had been observed in other small-scale experiments that herbivorous fish exert strong top-down effects on periphyton (e.g., *Wootton & Oemke, 1992*; *Flecker et al., 2002*; *Schneck, Schwarzbold & Melo, 2013*; *Martin et al., 2016*), and it was even shown that the small cyprinid species central stoneroller (*C. anomalum*) reduces periphyton on the mesohabitat scale (*Power, Matthews & Stewart, 1985*; *Stewart, 1987*; *Gelwick & Matthews, 1992*).

A possible explanation for the lower autotrophic periphyton biomass observed on fish exclosure tiles might be a strong indirect top-down effect of zoobenthivorous fish at the experimental site. This is unexpected because we assumed that direct top-down control by nase would be stronger than indirect top-down control over two trophic levels, especially as the biomass of nase was more than 100-fold higher than that of invertebrates at the experimental site. Unfortunately, there is uncertainty whether zoobenthivorous fish were completely excluded by the electrical fences on the tiles because we did not test the effectiveness of the electrical exclusion for small fish. However, other studies that used electrical exclosures at a similar intensity reported the exclusion of animals $\geq 1$ cm (*Pringle & Blake, 1994*; *Pringle & Hamazaki, 1997*; *Rosemond, Pringle & Ramirez, 1998*), which would have excluded all zoobenthivorous fish in our experiments. In addition, our results strongly indicate that benthic invertebrates might have been released from predation pressure, thereby increasing invertebrate grazing. This assumption is supported by our observation of a higher invertebrate grazer biomass in general and mayfly grazers in particular on exclosure tiles than on control tiles. If indeed invertebrate grazing was responsible for the observed results, then the indirect control of periphyton biomass by zoobenthivorous fish via invertebrate grazers was stronger than the direct top-down effect of herbivorous nase in our small-scale experiments.

In principle, the occurrence of a trophic cascade from fish over invertebrate grazers on periphyton seems likely because comparable effects have been observed under near-natural conditions in stream ecosystems for both zoobenthivorous fish (*Winkelmann et al., 2014*) and drift-feeding fish (*Huryn, 1998*; *Buria et al., 2010*; *Pagnucco, Remmal & Ricciardi, 2016*) and in many small-scale experiments (e.g., *Power, 1990b*; *Flecker & Townsend, 1994*; *Dahl, 1998*; *Kurle & Cardinale, 2011*). Moreover, effects of zoobenthivorous fish on the composition of the benthic invertebrate community have been observed on different experimental scales (large scale: *Winkelmann et al., 2007*; *Winkelmann et al., 2011*; *Worischka et al., 2014*; small scale: *Dahl, 1998*; *Shelton et al., 2016*). Grazing mayfly larvae and chironomids are important prey species for zoobenthivorous fish (*Copp, Spathari & Turmel, 2005*; *Ureche et al., 2010*; *Worischka et al., 2015*), which is in agreement with the strongest effects of fish exclusion on mayflies and chironomids in our experiments.

However, the strength of the observed effects did not correspond to the density of small zoobenthivorous fish (catch per m$^2$). Based only on the total density, the strongest effects would have been expected in experiment I with the highest density of zoobenthivorous fish (low nase density). However, our results showed the strongest effect in experiment III (significant decrease in Chl *a* concentrations and significant increase in benthic grazers), in which the total density of zoobenthivorous fish was lowest. One explanation for these results is that the strength of the predation pressure on the tiles might have been affected by a species shift within the zoobenthivorous fish community, owing to differences in their foraging behaviour. Stone loach (*B. barbatula*), whose stock was highest at high nase densities (experiment III), feeds in habitats with higher current velocities than other zoobenthivorous fish species (*Worischka et al., 2012*) and can therefore be expected to feed frequently on the experimental tiles. Another explanation for these results is that the strong effects are due to the relatively high density of the large omnivorous fish European chub (*S. cephalus*) and dace (*L. leuciscus*). Both species feed to a considerable proportion on benthic prey (*Vlach, Švátora & Dušek, 2013*) and might therefore have reduced benthic grazers on the tiles accessible to fish.

While the explanations stated above focused on top-down regulation, the higher Chl *a* concentration on accessible control tiles might also be the result of a stimulation of periphyton growth due to fish grazing, which represents an overcompensation of top-down regulation. During the first two weeks of experiment III, we frequently observed nase foraging on the control tiles and found highly visible traces of feeding. Such newly grazed patches, free from senescent algae and detritus, offer optimal growth conditions for new algae (*Lamberti & Resh, 1983*; *McCormick & Stevenson, 1989*), thus yielding comparatively more Chl *a*. This explanation is supported by the observation that while autotrophic biomass was significantly higher in the controls in experiments II and III, total biomass (AFDM), including heterotrophs and detritus other than living algae, was similar in all experiments. The presence of herbivorous fish on the accessible tiles might have caused increased removal of detritus (*Power, 1990a*; *Flecker, 1996*), allowing algae to increase their growth rate, thereby overcompensating fish grazing. The significantly lower variance of AFDM in the controls at high nase density (experiment III) could indicate that grazing by nase had a homogenizing effect on the spatial scale of the experimental site, leading to more similar total periphyton biomass in the controls, thereby supporting the second explanation.

Several methodical issues possibly led to an increase in variability and a lower observed effect of fish. First, annual and seasonal differences in the abundance and developmental stages of benthic invertebrates between the experiments cannot be ruled out completely. Unfortunately, we were not able to run the experiments in parallel and had to use a consecutive experimental design. An underestimation of indirect effects of zoobenthivorous fish might have resulted from the conservative estimation of the total grazer biomass. Especially for calculating the grazer biomass of Chironomidae, we used a small average proportion of plant food in their diet (20%; *Schmedtje & Colling, 1996*). Nevertheless, Chironomidae larvae are a diverse group that vary greatly in their feeding habits. Several taxa feed predominantly on algae (*Cummins, 1973*; *Pinder, 1992*; *Tarkowska-Kukuryk, 2013*) and

are able to exert strong top-down effects on periphyton (*Power, 1990b*; *Tarkowska-Kukuryk, 2013*). Assuming that mostly grazing chironomids settled on the tiles, it seems likely that grazing by chironomid larvae caused the lower Chl *a* concentration on the exclosure tiles in experiment II even though chironomid biomass did not significantly increase.

In addition, we suspect that especially in experiment I, bottom-up effects were likely more important than top-down effects because light intensity was highest and periphyton biomass was lowest for accessible and exclosure tiles of this experiment, while invertebrate grazer biomass was low. Higher light availability could have promoted fast periphyton growth, which leads to self-shading and ultimately to detachment of periphyton and thereby masks potential effects of fish exclusion (*Higgins, Hecky & Guildford, 2008*).

The use of artificial substrates, which might result in a different community structure of periphyton and invertebrates owing to uniform surface texture, size and colonization time (*Cattaneo & Amireault, 1992*), means our experimental results cannot be directly transferred to the situation in a real ecosystem. However, the relatively high proportion of filamentous green algae and cyanobacteria, especially the occurrence of attached taxa such as *Cladophora* spp., shows that the concrete tiles were a reasonably good facsimile and that colonization time was long enough to allow the development of mid- to late-successional stages of periphyton. Therefore, we think that although the periphyton assemblage on the tiles was not identical to that of natural substrates, it sufficiently represents the natural colonization of the river bed during summer.

Furthermore, the exact fish densities at the experimental site (20 m in length) were not known because fish moved within their natural home range. This uncertainty might have been overcome by using an enclosure design. However, such a design seems undesirable because it would have affected both abiotic conditions and fish behaviour. Because of the different habitat use of zoobenthivorous fish and nase, we chose different spatial scales for fish stock estimations. Zoobenthivorous fish are mostly stationary and small; we estimated their density in 60-m sections very close to the experimental sites. Nase, on the other hand, usually swim in shoals that move actively within defined home ranges (*Huber & Kirchhofer, 1998*). Therefore, we found it necessary to estimate nase stocks on a large scale (500 m) to reflect the potential grazing impact of nase at the experimental sites. Although we cannot rule out that nase changed their feeding places from time to time over the experimental periods, we expect that within the experimental period of more than two weeks, differences in day-to-day feeding areas were reasonably integrated over time.

Finally, the small spatial and temporal scale of our experiments does not allow us to draw a general conclusion from our experimental results on the possible top-down control of periphyton by nase in eutrophic rivers. Especially the effect of invertebrate grazing seems likely to be particularly strong on a small spatial scale but less relevant at larger scales (*Englund, 1997*; *Gil, Jiao & Osenberg, 2016*).

## CONCLUSIONS

Our results indicate that the active promotion of benthic grazing might be a possible tool to reduce eutrophication effects in rivers, but also highlight the complexity of top-down control in river food webs. In our small-scale experiments, cascading effects of

zoobenthivorous fish via invertebrate grazers might have been stronger than direct top-down effects of herbivorous nase. However, we cannot determine whether fish grazing or invertebrate grazing is more important on the ecosystem scale because the potential impact of herbivorous nase remained unclear, likely owing to the unspecific exclusion of fish in our experiment. To assess the top-down effects of herbivorous nase in eutrophic rivers, large-scale and long-term experiments that consider the impact of spatial and seasonal variability are needed.

## ACKNOWLEDGEMENTS

We thank Christian Sodemann and Manfred Fetthauer for technical advice and for valuable support during field work. We also thank all colleagues, students and volunteers of the local river protection association ARGE/Nister e.V. who helped with the electrofishing and field sampling. We are also grateful to Karen Brune for linguistic support. Furthermore, we thank the reviewers Mary Power and Kiran Liversage and the academic editor Richard Taylor for their thoughtful comments and suggestions, which greatly improved the manuscript.

### Funding

This work was supported by the Federal Office for Agriculture and Food of Germany (2813BM011) and the local environmental agency SGD Nord (Rhineland-Palatinate, Germany). The funders had no role in study design, data collection and analysis, decision to publish, or preparation of the manuscript.

### Grant Disclosures

The following grant information was disclosed by the authors:
Federal Office for Agriculture and Food of Germany: 2813BM011.
SGD Nord (Rhineland-Palatinate, Germany).

### Competing Interests

The authors declare there are no competing interests.

### Author Contributions

- Madlen Gerke performed the experiments, analyzed the data, wrote the paper, prepared figures and/or tables, reviewed drafts of the paper.
- Daniel Cob Chaves conceived and designed the experiments, performed the experiments, reviewed drafts of the paper.
- Marc Richter performed the experiments, reviewed drafts of the paper.
- Daniela Mewes performed the experiments, contributed reagents/materials/analysis tools, reviewed drafts of the paper.
- Jörg Schneider performed the experiments, contributed reagents/materials/analysis tools, reviewed drafts of the paper, organised and performed electrofishing-campaigns, contributed fish density data.

- Dirk Hübner contributed reagents/materials/analysis tools, reviewed drafts of the paper, organised and performed electrofishing-campaigns, contributed fish density data.
- Carola Winkelmann conceived and designed the experiments, analyzed the data, wrote the paper, reviewed drafts of the paper.

## Field Study Permissions

The following information was supplied relating to field study approvals (i.e., approving body and any reference numbers):

Electrofishing permit was obtained from the fisheries department of the local environmental agency SGD Nord (Rhineland-Palatinate, Germany).

## Data Availability

The raw data is available as Supplemental Files, and also at OwnCloud—University of Koblenz-Landau: https://owncloud.uni-koblenz-landau.de/owncloud/s/ASfpVRpiXvD3q1M.

## Supplemental Information

Supplemental information for this article can be found online at http://dx.doi.org/10.7717/peerj.4381#supplemental-information.

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

# PeerJ

**Higgins SN, Hecky RE, Guildford SJ. 2008.** The collapse of benthic macroalgal blooms in response to self-shading. *Freshwater Biology* **53**:2557–2572 DOI 10.1111/j.1365-2427.2008.02084.x.

**Hill WR, Knight AW. 1987.** Experimental analysis of the grazing interaction between a mayfly and stream algae. *Ecology* **68**:1955–1965 DOI 10.2307/1939886.

**Hillebrand H. 2009.** Meta-analysis of grazer control of periphyton biomass across aquatic ecosystems (1). *Journal of Phycology* **45**:798–806 DOI 10.1111/j.1529-8817.2009.00702.x.

**Hilton J, O'Hare M, Bowes MJ, Jones JI. 2006.** How green is my river? A new paradigm of eutrophication in rivers. *Science of the Total Environment* **365**:66–83 DOI 10.1016/j.scitotenv.2006.02.055.

**Holomuzki JR, Feminella JW, Power ME. 2010.** Biotic interactions in freshwater benthic habitats. *Journal of the North American Benthological Society* **29**:220–244 DOI 10.1899/08-044.1.

**Huber M, Kirchhofer A. 1998.** Radio telemetry as a tool to study habitat use of nase (*Chondrostoma nasus* L.) in medium-sized rivers. *Hydrobiologia* **371–372**:309–319 DOI 10.1023/A:1017005523302.

**Hübner D, Borchardt D, Fischer J. 2009.** Cascading effects of eutrophication on intragravel life stages of European grayling (*Thymallus thymallus* L.). *Advances in Limnology* **61**:205–224.

**Huryn AD. 1998.** Ecosystem-level evidence for top-down and bottom-up control of production in a grassland stream system. *Oecologia* **115**:173–183 DOI 10.1007/s004420050505.

**Ibisch RB, Seydell I, Borchardt D. 2009.** Influence of periphyton biomass dynamics on biological colmation processes in the hyporheic zone of a gravel bed river (River Lahn, Germany). *Advances in Limnology* **61**:87–104.

**Katano I, Doi H, Houki A, Isobe Y, Oishi T. 2007.** Changes in periphyton abundance and community structure with the dispersal of a caddisfly grazer, *Microsema quadriloba*. *Limnology* **8**:219–226 DOI 10.1007/s10201-007-0211-7.

**Kurle CM, Cardinale BJ. 2011.** Ecological factors associated with the strength of trophic cascades in streams. *Oikos* **120**:1897–1908 DOI 10.1111/j.1600-0706.2011.19465.x.

**Lamberti GA, Resh VH. 1983.** Stream periphyton and insect herbivores: an experimental study of grazing by a caddisfly population. *Ecology* **64**:1124–1135 DOI 10.2307/1937823.

**Mährlein M, Pätzig M, Brauns M, Dolman AM. 2016.** Length—mass relationships for lake macroinvertebrates corrected for back-transformation and preservation effects. *Hydrobiologia* **768**:37–50 DOI 10.1007/s10750-015-2526-4.

**Martin EC, Gido KB, Bello N, Dodds WK, Veach A. 2016.** Increasing fish taxonomic and functional richness affects ecosystem properties of small headwater prairie streams. *Freshwater Biology* **61**:887–898 DOI 10.1111/fwb.12752.

**McCormick PV, Stevenson RJ. 1989.** Effects of snail grazing on benthic algal community structure in different nutrient environments. *Journal of the North American Benthological Society* **8**:162–172 DOI 10.2307/1467634.

**Melcher AH, Lautsch E, Schmutz S. 2012.** Non-parametric methods—Tree and P-CFA—for the ecological evaluation and assessment of suitable aquatic habitats: a contribution to fish psychology. *Psychological Test and Assessment Modeling* **54**:293–306.

**Mewes D, Spielvogel S, Winkelmann C. 2017.** RNA/DNA ratio as a growth indicator of stream periphyton. *Freshwater Biology* **62**:807–818 DOI 10.1111/fwb.12903.

**Meyer E. 1989.** The relationship between body length parameters and dry mass in running water invertebrates. *Archiv Für Hydrobiologie* **117**:191–203.

**Moulton TP, De Souza ML, Silveira RML, Krsulović FAM. 2004.** Effects of ephemeropterans and shrimps on periphyton and sediments in a coastal stream (Atlantic forest, Rio de Janeiro, Brazil). *Journal of the North American Benthological Society* **23**:868–881 DOI 10.1899/0887-3593(2004)023<0868:EOEASO>2.0.CO;2.

**Oberdorff T, Guilbert E, Lucchetta JC. 1993.** Patterns of fish species richness in the Seine River basin, France. *Hydrobiologia* **259**:157–167 DOI 10.1007/BF00006595.

**Pagnucco KS, Remmal Y, Ricciardi A. 2016.** An invasive benthic fish magnifies trophic cascades and alters pelagic communities in an experimental freshwater system. *Freshwater Science* **35**:654–665 DOI 10.1086/685285.

**Peterson BJ, Deegan L, Helfrich J, Hobbie JE, Hullar M, Moller B, Ford TE, Hershey A, Hiltner A, Kipphut G, Lock MA, Fiebig DM, McKinley V, Miller MC, Vestal JR, Ventullo R, Volk G. 1993.** Biological responses of a tundra river to fertilization. *Ecology* **74**:653–672 DOI 10.2307/1940794.

**Pinder LCV. 1992.** Biology of epiphytic Chironomidae (Diptera:Nematocera) in chalk streams. *Hydrobiologia* **248**:39–51 DOI 10.1007/BF00008884.

**Power ME. 1990a.** Resource enhancement by indirect effects of grazers: armored catfish, algae, and sediment. *Ecology* **71**:897–904 DOI 10.2307/1937361.

**Power ME. 1990b.** Effects of fish in river food webs. *Science* **250**:811–814 DOI 10.1126/science.250.4982.811.

**Power ME, Dudley TL, Cooper SD. 1989.** Grazing catfish, fishing birds, and attached algae in a Panamanian stream. *Environmental Biology of Fishes* **26**:285–294 DOI 10.1007/BF00002465.

**Power ME, Matthews WJ, Stewart AJ. 1985.** Grazing minnows, piscivorous bass, and stream algae: dynamics of a strong interaction. *Ecology* **66**:1448–1456 DOI 10.2307/1938007.

**Pringle CM, Blake GA. 1994.** Quantitative effects of atyid shrimp (Decapoda: Atyidae) on the depositional environment in a tropical stream: use of electricity for experimental exclusion. *Canadian Journal of Fisheries and Aquatic Sciences* **51**:1443–1450 DOI 10.1139/f94-144.

**Pringle CM, Hamazaki T. 1997.** Effects of fishes on algal response to storms in a tropical stream. *Ecology* **78**:2432–2442 DOI 10.2307/2265904.

**R Development Core Team. 2016.** A language and environment for statistical computing. Vienna: R Foundation for Statistical Computing. *Available at https://r-project.org*.

**Reckendorfer W, Keckeis H, Tiitu V, Winkler G, Zornig H. 2001.** Diet shifts in 0+ nase, *Chondrostoma nasus*: size-specific differences and the effect of food. *Archiv fuer Hydrobiologie Supplement* **13512**:425–440.

**Ricker WE. 1975.** *Computation and interpretation of biological statistics of fish populations.* Ottawa: Department of the Environment, Fisheries and Marine Service, 382.

**Rosemond AD, Mulholland PJ, Brawley SH. 2000.** Seasonally shifting limitation of stream periphyton: response of algal populations and assemblage biomass and productivity to variation in light, nutrients, and herbivores. *Canadian Journal of Fisheries and Aquatic Sciences* **57**:66–75 DOI 10.1139/f99-181.

**Rosemond AD, Mulholland PJ, Elwood JW. 1993.** Top-down and bottom-up control of stream periphyton: effects of nutrients and herbivores. *Ecology* **74**:1264–1280 DOI 10.2307/1940495.

**Rosemond AD, Pringle CM, Ramirez A. 1998.** Macroconsumer effects on insect detritivores and detritus processing in a tropical stream. *Freshwater Biology* **39**:515–523 DOI 10.1046/j.1365-2427.1998.00301.x.

**Schmedtje U, Colling M. 1996.** Ökologische Typisierung der aquatischen Makrofauna. In: *Informationsberichte des Bayerischen Landesamtes für Wasserwirtschaft.* München: Bayerisches Landesamt für Wasserwirtschaft Bayerisches Landesamt für Wasserwirtschaft, 543.

**Schneck F, Schwarzbold A, Melo AS. 2013.** Substrate roughness, fish grazers, and mesohabitat type interact to determine algal biomass and sediment accrual in a high-altitude subtropical stream. *Hydrobiologia* **711**:165–173 DOI 10.1007/s10750-013-1477-x.

**Shapiro J, Wright DI. 1984.** Lake restoration by biomanipulation: Round Lake, Minnesota, the first two years. *Freshwater Biology* **14**:371–383 DOI 10.1111/j.1365-2427.1984.tb00161.x.

**Shelton JM, Samways MJ, Day JA, Woodford DJ. 2016.** Are native cyprinids or introduced salmonids stronger regulators of benthic invertebrates in South African headwater streams?: Impact of alien Trout in streams. *Austral Ecology* **41**:633–643 DOI 10.1111/aec.12352.

**Stewart AJ. 1987.** Responses of stream algae to grazing minnows and nutrients: a field test for interactions. *Oecologia* **72**:1–7 DOI 10.1007/BF00385036.

**Sturt MM, Jansen MAK, Harrison SSC. 2011.** Invertebrate grazing and riparian shade as controllers of nuisance algae in a eutrophic river. *Freshwater Biology* **56**:2580–2593 DOI 10.1111/j.1365-2427.2011.02684.x.

**Tarkowska-Kukuryk M. 2013.** Periphytic algae as food source for grazing chironomids in a shallow phytoplankton-dominated lake. *Limnologica—Ecology and Management of Inland Waters* **43**:254–264 DOI 10.1016/j.limno.2012.11.004.

**Ureche D, Ureche C, Nicoara M, Plavan G. 2010.** The role of macroinvertebrates in diets of fish in River Dambovita, Romania. *Verhandlungen des Internationalen Verein Limnologie* **30**:1582–1586.

**Vater M. 1997.** Age growth of the undermouth *Chondrostoma nasus* in the Slovak stretch of the Danube river. *Biologia* **52**:653–651.

**Vlach P, Švátora M, Dušek J. 2013.** The food niche overlap of five fish species in the Úpoř brook (Central Bohemia). *Knowledge and Management of Aquatic Ecosystems* **411(04)**:1–12 DOI 10.1051/kmae/2013070.

**Welch EB, Quinn JM, Hickey CW. 1992.** Periphyton biomass related to point-source nutrient enrichment in seven New Zealand streams. *Water Research* **26**:669–675 DOI 10.1016/0043-1354(92)90243-W.

**Winkelmann C, Hellmann C, Worischka S, Petzoldt T, Benndorf J. 2011.** Fish predation affects the structure of a benthic community. *Freshwater Biology* **56**:1030–1046 DOI 10.1111/j.1365-2427.2010.02543.x.

**Winkelmann C, Schneider J, Mewes D, Schmidt SI, Worischka S, Hellmann C, Benndorf J. 2014.** Top-down and bottom-up control of periphyton by benthivorous fish and light supply in two streams. *Freshwater Biology* **59**:803–818 DOI 10.1111/fwb.12305.

**Winkelmann C, Worischka S, Koop JHE, Benndorf J. 2007.** Predation effects of benthivorous fish on grazing and shredding macroinvertebrates in a detritus-based stream food web. *Limnologica—Ecology and Management of Inland Waters* **37**:121–128 DOI 10.1016/j.limno.2006.11.001.

**Wootton JT, Oemke MP. 1992.** Latitudinal differences in fish community trophic structure, and the role of fish herbivory in a Costa Rican stream. *Environmental Biology of Fishes* **35**:311–319 DOI 10.1007/BF00001899.

**Worischka S, Hellmann C, Berendonk TU, Winkelmann C. 2014.** Fish predation can induce mesohabitat-specific differences in food web structures in small stream ecosystems. *Aquatic Ecology* **48**:367–378 DOI 10.1007/s10452-014-9490-3.

**Worischka S, Koebsch C, Hellmann C, Winkelmann C. 2012.** Habitat overlap between predatory benthic fish and their invertebrate prey in streams: the relative influence of spatial and temporal factors on predation risk: habitat overlap and predation risk. *Freshwater Biology* **57**:2247–2261 DOI 10.1111/j.1365-2427.2012.02868.x.

**Worischka S, Schmidt SI, Hellmann C, Winkelmann C. 2015.** Selective predation by benthivorous fish on stream macroinvertebrates—The role of prey traits and prey abundance. *Limnologica—Ecology and Management of Inland Waters* **52**:41–50 DOI 10.1016/j.limno.2015.03.004.