# Peer review of "Benthic grazing in a eutrophic river: cascading effects of zoobenthivorous fish mask direct effects of herbivorous fish"

_PeerJ, doi:10.7717/peerj.4381_

## Round 0.1 · original submission · Major Revisions

Both reviewers have provided thoughtful and constructive comments on your MS. Please address these in your revision.

·

Basic reporting

More description of their study system and methods is needed. Most of these aspects should be relatively straight forward.

1. What is the drainage area of the river Nister? Drainage area is easily obtained from maps, and are much more quantitative than stream order.
2. What are the potential algivores (nace and major invertebrate grazers) in the system, how do they graze (browsing tips, clipping lawns, or scrubbing down to basal cells?) How do fish vs invertebrate grazing pressures change with substrate texture or stream velocity? In other words, tell us more about the natural history of the system, and which biota are likely to significantly affect algae.
3. I was confused about the scales over which densities were measured. Invertebrate densities were not described, and fishes were unconfined, so how do stock estimates from electroshocking relate to the densities of the nase and the invertebrate feeding fish over the scales that would influence standing crops of algae on the tiles.?
4. What taxa of algae or macrophytes were being grazed? Are they green algae, diatoms, cyanobacteria? Adnate or loosely attached? Are their natural substrates stony—were concrete blocks a good facsimile? When did the experiments occur within the seasonal algal succession and cycles of substrate colonization, vegetative growth, and sloughing that macro-algae or macrophytes show?

The English is adequate, but not always completely clear. There may be too much background information and there is certainly too little description of the actual biota or the study site or the spatial and time scales of natural processes (like fish foraging ranges or seasonal succession in stream biota) that matter.

Their raw data and authorizations for research all appear in order.

Experimental design

I think the design is ok if
1) the concrete tiles grow periphyton that is representative of what is on the natural river bed, but you should support this. One problem, in addition to surface texture and chemistry differences, is that if you place the tiles after natural colonization periods by algal propagules, you'll not grow the periphyton that may represent the natural flora of the stream bed.
2) you can explain what factors keep fishes at (or close enough to) the densities that you estimate with electro-shocking long enough to reflect their actual impact on the food chains that influence algal biomass over your experimental periods

Validity of the findings

The experiment you designed tests fish effects, not nase effects, on periphyton biomass on artificial substrates. You find that fish exclusion augments algal biomass, and infer (I suspect correctly) that invertivory may release algae from grazing invertebrates, which seem to have stronger effects than algivorous nase. If you can give us some clearer descriptions (and even better, some quantitative data) on the densities and grazing severity of grazing invertebrates versus nase, the paper would be a clearer contribution to our general understanding of which consumers can exert strong controls over producer biomass in rivers.

Additional comments

Top-down effects of herbivorous fish (Chondrostoma 1 nasus L.) in a eutrophic river (#21164)


The authors hypothesized that a fish they identify as the only obligate herbivorous fish in Europe, the cyprinid common nase (Chondrostoma nasus), may suppress periphyton growth in eutrophic rivers, as herbivorous fish are known to do in some midwestern North American and many tropical rivers. They tested their hypothesis with a field experiment in Rhineland-Palatinate, Germany in the river Nister, a large river enriched with human sewage and agrochemicals. They used electric fencing to exclude all fish from 40x40 cm2 concrete tiles placed on the river bed. They used natural densities that varied 8x over the first year, and in the second year, artificial stocking elevated nase densities 70x (but did the fish stay where they were stocked? See below.) The authors, however, did not find the effect they expected from nase exclusion. To their surprise, they found more algae where all fish were excluded, and concluded that benthic invertebrates were more important as grazers than fish. They offer a reasonable explanation, that invertebrates were more important grazers than nase in this system, and that excluding invertebrate-feeding fish, particularly loach, released invertebrates to graze down algae within fish exclosures. I think the large scale of the field study, the tests over a large range of free-swimming nase densities, and the lack of enclosure artifacts are all strengths of this ms, but a bit more clarification of scale and natural history is needed.

·

Basic reporting

The basic reporting was done well.

Experimental design

The experimental design was effective for testing effects of an electric field treatment on development of benthic river assemblages, and the methods were well described. But in some ways the design would not be effective for testing what was reported to have been tested, at least without further convincing by the authors (see below).

Validity of the findings

The findings of these experiments are valid concerning effects of excluding a subset of grazers (and predators) from early-successional benthic communities. The focus of the paper is different to what was actually tested though.

1. I am concerned about the uncertainty of what exactly was excluded by the electric field. The authors state that all fish were excluded, so the strong focus on just one species (nase) is not warranted. Nase may be expected to be the main grazer and have the largest effect, but they were not excluded separately from other fish, so the separate effect of nase is not known.

2. Also, the possibility of some invertebrates from being excluded was not mentioned. For example, some studies have found large impacts of electricity not only on fish but also invertebrates (Brown et al. (2000) J. N. Am. Benthol. Soc. 19:176-185). Overall, the experiment can not tell if effects on periphyton are caused by exclusion of all (or some) fish, some invertebrates, or changed predation of the fish on the invertebrates. All it can answer is – what are the effects of species sensitive to an electric field on benthic assemblages? I think it is necessary to do some additional experiments to determine if any invertebrates are or are not independently affected by electricity. Or possibly the authors can find some past studies (including similar invertebrate species) that show an electric field of the same strength used in this study does not affect their abundance or behavior.

3. It is good that three experiments were done, but they are done at different times and the contexts of the experiments will differ in many ways besides just differences in nase abundance among those times. I think it is all right that the authors do include some speculation about how differences in nase abundance may contribute to differences in results among the experiments, but differences of nase density should not be overly emphasised (e.g. line 260, 314). The emphasis should be on differences among separate experiments that may differ in many ways including, but not limited to, nase density.

4. The tiles were left in the river for a total of 32-33 days, so I would think these will still be early-successional communities that have developed. If so, then the results are only applicable to these kinds of assemblages, which would need to be stated. For example, late-successional assemblages may have benthic algae that are more resistant to grazing and the results would be very different. If the natural benthos is dominated by late-successional assemblages then the results may have limited relevance.

5. The experiment did not include measurements of naturally-occurring river substrata (e.g. boulders or flat rock) so it is unknown how the colonisation of the tiles may differ to colonisation of natural substrata. For example, the surface textures may be different and affect periphyton growth. Possible differences between the natural and artificial substrata need to be discussed. If the authors have any data on what the algal and macroinvertebrate assemblages were like on surrounding natural rock surfaces they should consider including these data.

6. The opening of the abstract and introduction is about eutrophication, but the experiments are only indirectly relatable this process. I think the beginning of the introduction and abstract should be changed to be related to effects of fish, then the possible consequences for eutrophication can be mentioned. As it is, readers interested mostly in eutrophication may be confused when they realise this study it more about fish affecting the benthos.

Additional comments

1. Line 40: This should be reworded because the only comparisons of periphtyon biomass showed that there was no significant different among treatments (Line 266). The difference was in chlorophyll concentrations, which may or may not be related to biomass.
2. Line 154 – could you please include details of these preliminary experiments, perhaps as supplementary material? I would like to know details such as 1) how observations were made (were the authors certain that the nase were completely excluded during the 24hrs?), 2) were the fish starved before the trial and what kind of food were they provided near the electrified area? (if the fish were not actively foraging then they would not have any reason to go near the baited electrified area), 3) was a control area included that was not electrified in this trial? Could the authors also please include discussion about how they did not test the effectiveness of the electrified treatment on any of the many other species of fish that co-occurred with nase and which also will have affected the benthic assemblages.
3. Line 150 – did the fence chargers create a consent electric field, or one that pulsed on and off?
4. line 162 – I had trouble understanding exactly how many treatments were used. If this line means that there were 9 control tiles and 9 electrified tiles in each experiment, could the authors please make it a little clearer.
5. Line 236 – I think there is a error in this sentence
6. Line 245 – change to “weighed”
7. Line 248-249 - Can you please explain what the multiple comparisons were that needed correcting for?
8. Line 265-266 – any non-significant effect on AFDW should be mentioned in the abstract along with the significant effect on chlorophyll.
9. Line 288 – change to “There was no evidence that the environmental factors water depth, light and current velocity affected the experimental results.”
10. Line 329 - 331– the potential influence of invertebrate grazers is something that could easily have been predicted before the start of the experiment, so I think it needs to be incorporated into the hypotheses. Overall, the experiment would be made much more valuable if some more data on separate effects of invertebrates were included.
11. Line 396 – change “special” to “spatial”
12. Figure 2 – I think the Asterisks are not meant to be there.

---

## Round 0.2 · Minor Revisions

Thank you for revising your MS. You've done a good job of addressing most parts of the original reviews, but I agree with the reviewer of the revised MS that a few more changes are required before acceptance. In addition to the reviewer’s comments, could you please address the following (line #s from the Word doc):

Line 38: delete “also”

In the Abstract mention that biomasses of grazing invertebrates were very low relative to base.

Line 484: The final part of the following sentence is too negative, and is contradicted by the rest of the paragraph. “The use of artificial substrates, which might result in a different community structure of periphyton and invertebrates owing to uniform surface texture, size and colonization time (Cattaneo & Amireault, 1992), does not allow the results of our experiments to be transferred to the situation in a real ecosystem.” Perhaps change the ending to something like “, means our experimental results cannot automatically be extrapolated to the field.”

·

Basic reporting

no comment

Experimental design

no comment

Validity of the findings

no comment

Additional comments

I have provided some additional minor changes. The line numbers below correspond to the line numbers on the “tracked changes manuscript”, which had line numbers that were slightly different to the reviewing PDF for some reason.

Title: I would consider changing the title and expanding it with a bit more detail of the findings. The current one does not really reflect what was found in the study because there was no top-down effects of nase, while the important effect was likely from other fish eating the invertebrate grazers.

Line 23-24: This sentence in the abstract seems out of place and the mention of generation times being mismatched will confuse readers. Maybe change it to something like “Although benthic invertebrate grazers reduce the growth of periphyton, this is highly context dependent”.

Line 40: I may be wrong but I would guess that “benthivorous” means species that eat anything from the benthos, and this would include not only invertebrates but also algae, so maybe change to “predators of benthic invertebrates”. This may need changing throughout the MS.

Line 44-50: Starting the introduction with mention of specifically invertebrate grazing is a bit confusing, especially given invertebrates are not mentioned in the title. Perhaps you could merge the 1st and 2nd paragraphs so you mention both types of grazing (invert. and fish) together.

Line 82-83: I still find this ambiguous. It might mean there was one set of tiles per experiment, including 9 tiles some of which were electrified and some weren’t, or it could mean that each experiment had two sets, including 18 tiles altogether, some of which were electrified and some weren’t. This is an important piece of information, so try to be very clear, e.g. “Each separate experiment had a total of 18 tiles; half the tiles were electrified to exclude fish, and the other half were non-electrified to allow fish access”

Line 319: are you certain that Standard Deviation here is meant to be a percentage?

Line 335-336: Is there any reason this is in square brackets? Also, why is SD shown for the tiles but not the stone. Also, check whether you are presenting here Standard Deviation or Standard Error.

Line 409: change “we have no proof” to “there is uncertainty whether”

Line 413: Not sure what you mean by “quantitatively” here

Line 486: add “directly” before “transferred”

---

## Round 0.3 · accepted · Accept

Thank you for making the requested revisions.